# Backplay: 'Man muss immer umkehren'

**Cinjon Resnick**[*]
NYU
cinjon@nyu.edu

**Roberta Raileanu**[*]
NYU
rr3009@nyu.edu

**Sanyam Kapoor**
NYU
sanyam@nyu.edu

**Alexander Peysakhovich**
FAIR
alexpeys@fb.com

**Kyunghyun Cho**
NYU, FAIR
kyunghyun.cho@nyu.edu

**Joan Bruna**
NYU, FAIR
bruna@cims.nyu.edu

## ABSTRACT

Model-free reinforcement learning (RL) requires a large number of trials to learn a good policy, especially in environments with sparse rewards. We explore a method to improve the sample efficiency when we have access to demonstrations. Our approach, Backplay, uses a single demonstration to construct a curriculum for a given task. Rather than starting each training episode in the environment's fixed initial state, we start the agent near the end of the demonstration and move the starting point backwards during the course of training until we reach the initial state. Our contributions are that we analytically characterize the types of environments where Backplay can improve training speed, demonstrate the effectiveness of Backplay both in large grid worlds and a complex four player zero-sum game (Pommerman), and show that Backplay compares favorably to other competitive methods known to improve sample efficiency. This includes reward shaping, behavioral cloning, and reverse curriculum generation.

## 1 INTRODUCTION

An important goal of AI research is to construct agents that can learn well in new environments (Levine et al., 2016). An increasingly popular paradigm for this task is deep reinforcement learning (deep RL, Silver et al. (2016); Moravcík et al. (2017); Silver et al. (2017)). However, training an RL agent can take a very long time, particularly in environments with sparse rewards. In these settings, the agent typically requires a large number of episodes to stumble upon positive rewards and learn even a moderately effective policy that can then be refined. This is often resolved via hand-engineering a dense reward function. Such reward shaping, while effective, can also change the set of optimal policies and have unintended side effects (Ng et al., 1999; Clark & Amodei, 2016).

We consider an alternative technique for accelerating RL in sparse reward settings. The idea is to create a curriculum for the agent via reversing a single trajectory (i.e. state sequence) of reasonably good, but not necessarily optimal, behavior. We start our agent at the end of a demonstration and let it learn a policy in this easier setup. We then move the starting point backward until the agent is training only on the initial state of the task. We call this technique **Backplay**.

Our contributions are threefold:

1. We characterize analytically and qualitatively which environments Backplay will aid.

2. We demonstrate Backplay's effectiveness on both a grid world task (to gain intuition) as well as the four player stochastic zero-sum game Pommerman (MultiAgentLearning, 2018).

3. We empirically show that Backplay compares favorably to other methods that improve sample complexity.

Besides requiring vastly fewer number of samples to learn a good policy, an agent trained with Backplay can outperform its demonstrator and even learn an optimal policy following a sub-optimal

---

[*]These two authors contributed equally

demonstration. Our experiments further show Backplay's strong performance relative to reward shaping (involves hand tuning reward functions), behavioral cloning (not intended for use with sub-optimal experts), and other forms of automatic curriculum generation (Florensa et al. (2017), requires a reversable environment).

## 2 RELATED WORK

The most related work to ours is a blog post describing a method similar to Backplay used to obtain state-of-the-art performance on the challenging Atari game Montezuma's Revenge (Salimans & Chen, 2018). This work was independent of and concurrent to our own. In addition to reporting results on a different, complex stochastic multi-agent environment, we provide an analytic characterization of the method as well as an in depth discussion of what kinds of environments a practitioner can expect Backplay to out or underperform other existing methods.

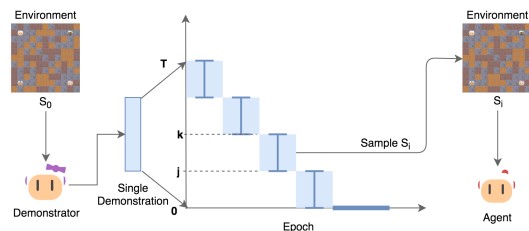

Figure 1. Backplay: We first collect a demonstration, from which we build a curriculum over the states. We then sample a state according to that curriculum and initialize our agent accordingly.

A popular method for improving RL with access to expert demonstrations is behavioral cloning/imitation learning. These methods explicitly encourage the learned policy to mimic an expert policy (Bain & Sommut, 1999; Ross et al., 2011; Daumé et al., 2009; Zhang & Cho, 2016; Laskey et al., 2016; Nair et al., 2017; Hester et al., 2017; Ho & Ermon, 2016; Aytar et al., 2018; Lerer & Peysakhovich, 2018; Peng et al., 2018). Imitation learning requires access to both state and expert actions (whereas Backplay only requires states) and is designed to copy an expert, thus it cannot, without further adjustments (e.g. as proposed by Gao et al. (2018)), surpass a suboptimal expert. We discuss the pros and cons of an imitation learning + adjustment vs. a Backplay-based approach in the main analysis section.

Other algorithms (Ranzato et al., 2015; Li et al., 2016; Das et al., 2017a;b), primarily in dialog, use a Backplay-like curriculum, albeit they utilize behavioral cloning for the first part of the trajectory. This is a major difference as we show that for many classes of problems, we *only* need to change the initial state distribution and do not see any gains from warm-starting with imitation learning. Backplay is more similar to Conservative Policy Iteration (Kakade & Langford, 2002), a theoretical paper which presents an algorithm designed to operate with an explicit restart distribution.

Also related to Backplay is the method of automatic reverse curriculum generation Florensa et al. (2017). These approaches assumes that the final goal state is known and that the environment is both resettable and reversible. The curricula are generated by taking random walks in the state space to generate starting states or by taking random actions starting at the goal state McAleer et al. (2018). These methods do not require an explicit 'good enough' demonstration as Backplay does. However, they require the environment to be reversible, an assumption that doesn't hold in many realistic tasks such as a robot manipulating breakable objects or complex video games such as Starcraft. In addition, they may fare poorly when random walks reach parts of the state space that are not actually relevant for learning a good policy. Thus, whether a practitioner wants to generate curricula from a trajectory or a random walk depends on the environment's properties. We discuss this in more detail in our analysis section and show empirical results suggesting that Backplay is superior.

Hosu & Rebedea (2016) use uniformly random states of an expert demonstration as starting states for a policy. Like Backplay, they show that using a single loss function to learn a policy from both demonstrations and rewards can outperform the demonstrator and is robust to sub-optimal demonstrations. However, they do not impose a curriculum over the demonstration and are equivalent to the Uniform baseline in our experiments.

Zhu et al. (2018) use a curriculum but manually tune it for each 'stage' of the environment. Within each stage, they use what we call Uniform training, which fails in our most challenging environments.

Goyal et al. (2018) and Edwards et al. (2018) simultaneously proposed the use of a learned backtracking model to generate traces that lead to high value states. Their methods rely on either having the agent visit high reward states or learning a model of the environment capable of generating the states. Both of these are challenging in environments in which the dynamics near starting states are very different from those near goal states.

Finally, Ivanovic et al. (2018) use a known (approximate) dynamics model to create a backwards curriculum for continuous control tasks. Their approach requires a physical prior which is not always available and often not applicable in multi-agent scenarios. In contrast, Backplay automatically creates a curriculum fit for any resettable environment with accompanying demonstrations.

## 3 BACKPLAY

Consider the standard formalism of a single agent Markov Decision Process (MDP) defined by a set of states $\mathcal{S}$, a set of actions $\mathcal{A}$, and a transition function $\mathcal{T} : \mathcal{S} \times \mathcal{A} \to \mathcal{S}$ which gives the probability distribution of the next state given a current state and action. If $\mathcal{P}(\mathcal{A})$ denotes the space of probability distributions over actions, the agent chooses actions by sampling from a stochastic policy $\pi : \mathcal{S} \to \mathcal{P}(\mathcal{A})$, and receives reward $r : \mathcal{S} \times \mathcal{A} \to \mathbb{R}$ at every time step. The agent's goal is to construct a policy which minimizes its discounted expected return $R_t = \mathbb{E}\left[\sum_{k=0}^{\infty} \gamma^k r_{t+k+1}\right]$ where $r_t$ is the reward at time $t$ and $\gamma \in [0, 1]$ is the discount factor, and the expectation is taken with respect to both the policy and the environment.

The final component of an MDP is the distribution of initial starting states $s_0$. The key idea in Backplay is that we do not initialize the MDP in only a fixed $s_0$. Instead, we assume access to a demonstration which reaches a sequence of states $\{s_0^d, s_1^d, \ldots, s_T^d\}$. For each training episode, we uniformly sample starting states from the sub-sequence $\{s_{T-k}^d, s_{T-k+1}^d, \ldots, s_{T-j}^d\}$ for some window $[j, k]$. Note that this training regime requires the ability to reset the environment to any state. As training continues, we 'advance' the window according to a curriculum by increasing the values of $j$ and $k$ until we are training on the initial state in every episode ($j = k = T$). In this manner, our hyperparameters for Backplay are the windows and the training epochs at which we advance them.

### 3.1 QUANTITATIVE ANALYSIS

Next, we consider a simple environment in which we can analytically show that Backplay will improve the sample-efficiency of RL training.

Consider a connected, undirected graph $G = (V, E)$. An agent is placed at a fixed node $v_0 \in V$ and moves to neighboring nodes at a negative reward of $-1$ for each step. Its goal is to reach a target node $v_* \in V$, terminating the episode. This corresponds to an MDP $\mathcal{M} = (\mathcal{S}, \mathcal{A}, P, R)$ with state space $\mathcal{S} \sim V$, action space $\mathcal{A} \sim E$, deterministic transition kernel corresponding to

$$P(s_{t+1} = j \mid s_t = i, a_t = (l, k)) = \delta(j = k)\delta(l = i) + \delta(j = i)\delta(l \neq i)$$

and uniform reward $R(s_t, a_t) = -1$ for $s_{t+1} \neq v_*$. Assume that $\pi$ is a fixed policy on $\mathcal{M}$, such that the underlying Markov chain is irreducible and aperiodic:

$$K_\pi(s', s) = \sum_{a \in \mathcal{A}} P(s_{t+1} = s' \mid s_t = s, a_t = a)\pi(a \mid s)$$

$K_\pi$ has a single absorbing state at $s = s_* := v_*$. Our goal is to estimate the value function of this policy, equivalent to the expected first-passage time of the Markov chain $K_\pi$ from $s_0 = s$ to $s_*$:

$$V_\pi(s) = \mathbb{E}\tau(s, s_*) = \mathbb{E}\min\{j \geq 0 \; ; \; s.t. \, s_j = s_*, s_0 = s, s_{i+1} \sim K_\pi(\cdot, s_i)\} \, .$$

We consider a special tractable case where the value function can be well approximated by looking at the distance of a state from the goal state. Formally:

**Assumption 1.** *For all $\theta$, $V_\theta(s) = V_\theta(s')$ whenever $dist_G(s', s_*) = dist_G(s, s_*)$.*

We wish to analyze the process which is the projection of our Markov policy $\pi$ only in terms of the distance $z_t$. However, now the transition probabilities will not only be a function of only $z_t$ and so

this projection will be non-Markovian. Its Markov approximation $\bar{z}_t$ is defined as the Markov chain $\overline{K}$ given by the expected transition probabilities

$$
\begin{aligned}
\Pr(\bar{z}_{t+1} = l - 1 | \bar{z}_t = l) &= \alpha_l := \Pr_\mu(z_{t+1} = l - 1 \mid z_t = l) \,, & (1) \\
\Pr(\bar{z}_{t+1} = l | \bar{z}_t = l) &= \beta_l := \Pr_\mu(z_{t+1} = l \mid z_t = l) \,, \; l = 0 \dots M & (2)
\end{aligned}
$$

and thus $\Pr(\bar{z}_{t+1} = l + 1 | \bar{z}_t = l) = 1 - \alpha_l - \beta_l = \Pr_\mu(z_{t+1} = l + 1 \mid z_t = l)$ under the stationary distribution $\mu$ of $K_\pi$.

**Assumption 2.** *The projected process is well described by its Markovian approximation.*

Though these assumptions are relatively strong, they makes the analysis of Backplay in this graph complex but analytically tractable. Given a demonstration $\mathbf{d} = (d_0 = s_0, \dots, d_L = s_*)$, $d_l \in \mathcal{S}$ we will perform Backplay

**Theorem 1.** *When assumptions 1 and 2 hold, the sample complexity gains from using Backplay rather than standard RL are exponential in the diameter of the graph. $O(\frac{M^2}{m}\alpha^{-m})$ vs $\Omega(M\alpha^{-M/2})$.*

*Proof.* we sample it using a step size $m > 0$ (such that $j = 0 \bmod m$, where $j$ is defined in Section 3) to obtain $\bar{d}_l := \text{dist}_G(d_{L-ml}, s_*)$, $l = 0, 1, \dots, \frac{L}{m}$, which satisfies $\bar{d}_l \leq lm$ for all $l$.

For fixed $l$, we initialize the chain $\overline{K}$ at $\bar{d}_l$: $\bar{z}_0 = \bar{d}_l$. Since $Pr(\bar{z}_m = 0) \geq \prod_{j=0}^{m-1} \alpha_j := \gamma_{0,m}$, after $O(\gamma_{0,m}^{-1})$ trials of length $\leq M$, we will reach the absorbing state and finally have a signal-carrying update for the Q-function at the originating state.

We can consequently merge that state into the absorbing state and reduce the length of the chain by one. Repeat the argument $m$ times so that after $O(\sum_{j=0}^{m} \gamma_{j,m}^{-1}) = O(m\gamma_{0,m}^{-1})$ trials, the Q-function is updated at $\bar{z}_0$. Repeat at Backplay steps $lm$, $l = 1, \dots \frac{M}{m}$, and we reach a sample complexity of

$$
T_m = \sum_{k=0}^{\frac{M}{m}-1} O\left( M \sum_{j=0}^{m} \gamma_{km,(k+1)m-j}^{-1} \right) .
$$

In the case where $\alpha_l = \alpha$ for all $l$, we obtain $\gamma_{km,(k+1)m-j}^{-1} = \gamma_{0,m-j} = \alpha^{-m+j}$ and therefore $T_m = O\left( \frac{M^2(1-\alpha^{m+1})}{m(1-\alpha)}\alpha^{-m} \right)$, where **the important term is the rate** $\frac{M^2}{m}\alpha^{-m}$.

On the other hand, Hong & Zhou (2017) shows that the first-passage time $\tau(0, M)$ in a skip-free finite Markov chain of $M$ states with a single absorbing state is a random variable whose moment-generating function $\varphi(s) = \mathbb{E}s^{\tau(s,s_*)}$ is given by

$$
\varphi(s) = \prod_{j=1}^{M} \frac{(1-\lambda_j)s}{1-\lambda_j s} \,, \tag{3}
$$

where $\lambda_1, \dots, \lambda_M$ are the non-unit eigenvalues of the transition probability matrix. It follows that $\mathbb{E}\tau(0, M) = \varphi'(1) = \sum_{j=1}^{M} \frac{1}{1-\lambda_j} \approx (1 - \lambda_1)^{-1}$, which corresponds to the reciprocal spectral gap.[1] Chen & Saloff-Coste (2013) further shows that this reciprocal spectral gap is $\Omega(\alpha^{-M/2})$ in our case, and therefore the model without Backplay will on average take $T_M = \Omega(\alpha^{-M/2})$ trials to reach the absorbing state and receive information. □

We can analyze the uniform strategy similarly. The probability that a trajectory initialized at one of the uniform samples will reach the absorbing state is lower bounded by

$$
\sum_{j=1}^{M} \alpha^j P(\bar{z}_0 = j) = \frac{1}{M} \sum_{j=1}^{M} \alpha^j = \frac{\alpha - \alpha^{M+1}}{M(1-\alpha)} \,, \tag{4}
$$

which is approximately $\frac{\alpha}{M}$ when $\alpha$ is small, leading to a sample complexity of $O(M^2\alpha^{-1})$ to update the value function at the originating state, and $O(M^3\alpha^{-1})$ at the starting state. Comparing this rate

---

[1]The *spectral gap* of a Markov Chain is the difference between the first and second largest eigenvalue magnitudes of its transition probability matrix. It controls its mixing time among many other properties.

to Backplay with $m = 1$, observe that the uniform strategy is slower by a factor of $M$ (and one can verify that the same is true for generic step size $m$ by imagining that we first sampled a window of size $m$ and then sub-sampled our state from that window), suggesting that it loses efficiency on environments with large diameter.

The preceding analysis suggests that a full characterization of Backplay is a fruitful direction for reinforcement and imitation learning theory, albeit beyond the scope of this paper.

## 3.2 QUALITATIVE ANALYSIS

We now provide intuition regarding the conditions under which Backplay can improve sample-efficiency or lead to a better policy than that of the demonstrator. In addition, we discuss the differences between Backplay and other methods of reducing sample complexity for deep RL as well as when practitioners would choose to use one or the other.

Figure 2 contains three grid worlds. In each, the agent begins at $s_0$, receives $+1$ when it reaches $s_*$, and otherwise incurs a per step cost. They each pose a challenge to model free RL and highlight advantages and disadvantages of Backplay compared to other approaches like behavioral cloning (BC, Bain & Sommut (1999)), generative adversarial imitation learning (GAIL, Ho & Ermon (2016)), and reverse curriculum generation (RCG, Florensa et al. (2017)). See Table 1 for a direct comparison of these algorithms.

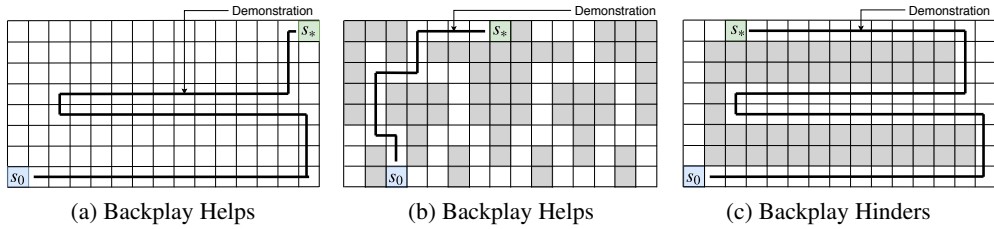

| (a) Backplay Helps | (b) Backplay Helps | (c) Backplay Hinders |

Figure 2. Three environments illustrating when Backplay can help or hinder learning an optimal policy. Backplay is expected to learn faster than standard RL on the first and second mazes, but perform worse on the third maze.

The left grid world shows a sparse reward environment in which Backplay can decrease the requisite training time compared to standard RL. Using the sub-optimal demonstration will position the Backplay agent close to high value states. In addition, the agent will likely surpass the expert policy because, unlike in BC approaches, Backplay does not encourage the agent to imitate expert actions. Rather, the curriculum forces the agent to first explore states with large associated value and, consequently, estimating the value function suffers less from the curse of dimensionality. And finally, we expect it to also surpass results from RCG because random movements from the goal state will progress very haphazardly in such an open world.

The middle grid illustrates a maze with bottlenecks. Backplay will vastly decrease the exploration time relative to standard RL, because the agent will be less prone to exploring the errant right side chamber where it can get lost if it traverses to the right instead of going up to the goal. If we used BC, then the agent will sufficiently learn the optimal policy, however it will suffer when placed in nearby spots on the grid as they will be out of distribution; Backplay-trained agents will not have this problem as they also explore nearby states. When an RCG agent reaches the first fork, it has an even chance of exploring the right side of the grid and wasting lots of sample time.

On the rightmost grid world, while Backplay is still likely to surpass its demonstrator, it will have trouble doing better than standard RL because the latter always starts at $s_0$ and consequently is more likely to stumble upon the optimal solution of going up and right. In contrast, Backplay will spend the dominant amount of its early training starting in states in the sub-optimal demonstration. Note that BC will be worse off than Backplay because by learning the demonstration, it will follow the trajectory into the basin instead of going up the right side. Finally, observe that RCG will likely outperform here given that it has a high chance of discovering the left side shortcut and, if not, it would more likely discover the right side shortcut than be trapped in the basin.

In summary, Backplay is not a universal strategy to improve sample complexity. Even in the navigation setting, if the task randomizes the initial state $s_0$, a single demonstration trajectory does not generally improve the coverage of the state-space outside an exponentially small region around said trajectory. For example, imagine a binary tree and a navigation task that starts at a random leaf and needs to reach the root. A single expert trajectory will be disjoint from half of the state space (because the root is absorbing), thus providing no sample complexity gains on average.

| Method | Requirements | Main Idea | Main Weakness |
|---|---|---|---|
| BC | (State, Action) pairs from expert trajectory. | Learn policy that imitates the expert demonstration. | Sub-optimal expert can yield a very poor learned policy. |
| GAIL | (State, Action) pairs from expert trajectory. | Learn a policy that matches the distribution of expert trajectory pairs. | Difficult to tune; Requires more world interactions; Can perform worse than BC. |
| RCG | Reversable transition function of environment; Resettable environment. | Take random walks from goal state to build curriculum of initial starting states. | Complexity may increase if random walks reach parts of state space irrelevant to a good policy. |
| Backplay | State sequence from a 'good enough' trajectory; Resettable environment. | Sample starting state from given trajectory by walking backward along trajectory. | If states in 'good enough' trajectory are not optimal, then can slow learning the *optimal* policy. |

Table 1: Comparison of Backplay with related work: Behavioral Cloning (BC), Generative Adversarial Imitation Learning (GAIL), and Reverse Curriculum Generation (RCG).

## 4  EXPERIMENTS

We now move to evaluating Backplay empirically in two environments: a grid world maze and a four-player free-for-all game. The questions we study across both environments are the following:

- Is Backplay more efficient than training an RL agent from scratch?
- How does the quality of the given demonstration affect the effectiveness of Backplay?
- Can Backplay agents surpass the demonstrator when it is non-optimal?
- Can Backplay agents generalize?

### 4.1  TRAINING DETAILS

We compare several training regimes. The first is **Backplay**, which uses the Backplay algorithm corresponding to a particular sequence of windows and epochs as specified in A.1. The second, **Standard** is vanilla model-free RL with the agent always starting at the initial state $s_0$. The last, **Uniform**, is an ablation that considers how important is the curriculum aspect of Backplay by sampling initial states randomly from the entire demonstration. In all these regimes, we use Proximal Policy Optimization (PPO, Schulman et al. (2017)) to train an agent with policy and value functions parameterized by convolutional neural networks.

On the Maze environment (detailed below), we also ran comparisons against **Behavioral Cloning** and **Reverse Curriculum Generation**. We chose BC over GAIL (Ho & Ermon, 2016) for three reasons. First, GAIL requires careful hyperparameter tuning and is thus difficult to train. Second, GAIL requires more environment interactions. And third, GAIL has recently been shown to perform significantly worse than BC (Behbahani et al., 2018).

Training details and network architectures for all the environments can be found in A.3 and A.6, while A.9 contains empirical observations for using Backplay in practice.

### 4.2  MAZE

We generated mazes of size $24 \times 24$ with 120 randomly placed walls, a random start position, and a random goal position. We then used A* to generate trajectories. These included both Optimal demonstrations (true shortest path) and N-Optimal demonstrations (N steps longer than the shortest path). More details on this setup are given in A.2.

| Algorithm | Map Set | % Optimal | % 0-5 Optimal | Avg Suboptimality | Std Suboptimality |
|---|---|---|---|---|---|
| Standard | All | 0 | 0 | N/A | N/A |
| Uniform | Optimal | 27 | 91 | 8.26 | 32.92 |
| Uniform | 5-Optimal | 51 | 98 | 2.04 | 17.39 |
| Uniform | 10-Optimal | 49 | 98 | 2.04 | 16.79 |
| Florensa | Optimal | 20 | 51 | 63.36 | 78.08 |
| Florensa | 5-Optimal | 48 | 77 | 25.44 | 56.89 |
| Florensa | 10-Optimal | 49 | 69 | 39.75 | 70.92 |
| Backplay | Optimal | **31** | **100** | 0.64 | **4.96** |
| Backplay | 5-Optimal | 37 | 94 | 7.94 | 33.35 |
| Backplay | 10-Optimal | **54** | 99 | **0.37** | 3.49 |

Table 2: Results after 2000 epochs on 100 mazes. Note that for Backplay and Uniform, the Map Set is also the type of demonstrator, where N-optimal has demonstrations N steps longer than the shortest path. From left to right, the table shows: the percentage of mazes on which the agent optimally reaches the goal, percentage on which it reaches in at most five steps more than optimal, and the average and standard deviation of extra steps over optimal. Both Backplay and Uniform succeed on almost all mazes and, importantly, can outperform the experts' demonstrations. On the other hand, Standard does not learn a useful policy and Florensa fails to learn more than $50 - 70\%$ of the maps, which is why its sub-optimality mean and std is so high. Results for Backplay were generally consistent across all seeds. For others, we report their best score. See A.4 for further details.

Our model receives as input four $24 \times 24$ binary maps. They contain ones at the positions of, respectively, the agent, the goal, passages, and walls. It outputs one of five options: Pass, Up, Down, Left, or Right. The game ends when the agent has reached its goal or after a maximum of 200 steps, whereupon the agent receives reward of +1 if it reaches the goal and a per step penalty of -0.03.

**Backplay, Uniform, Standard**. Table 2 shows that Standard has immense trouble learning in this sparse reward environment while both Backplay and Uniform find an optimal path approximately 30-50% of the time and a path within five of the optimal path almost always. Thus, in this environment, demonstrations of even sub-optimal experts are extremely useful, while the curriculum created by Backplay is not necessary. That curriculum does, however, aid convergence speed (A.4). We will see in 4.3 that the curriculum becomes vital as the environment increases in complexity.

**Behavioral Cloning.** We trained an agent using behavioral cloning from the same trajectories as the ones used for Backplay. While the agent learns to perfectly imitate those trajectories, we had immense difficult doing better than the expert. All of our attempts to use reward signal to improve the agent's performance (over that of the behaviorally cloned agent) were unsuccessful, even after incorporating tricks in the literature such as those found in Schmitt et al. (2018). One possible reason is that the agent has no information about states outside of the demonstration trajectory.

**Reverse Curriculum Generation.** We also compared Backplay to the method proposed by Florensa et al. (2017). As illustrated in Table 2 and Section A.4, Florensa agents perform significantly worse with higher sample complexity variance compared to Backplay (or Uniform). This suggests that having demonstrations helps inordinately. Further details can be found in A.3.

### 4.3 POMMERMAN

Pommerman is a stochastic environment (Resnick et al., 2018) based on the classic console game Bomberman and will be a competition at NeurIPS 2018. It is played on an 11x11 grid where on every turn, each of four agents either move in a cardinal direction, pass, or lay a bomb. The agents begin fenced in their own area by two different types of walls - rigid and wooden. The former are indestructible while bombs destroy the latter. Upon blowing up wooden walls, there is a uniform chance at yielding one of three power-ups: an extra bomb, an extra unit of range in the agent's bombs, or the ability to kick bombs. The maps are designed randomly, albeit there is always a guaranteed path between any two agents. For a visual aid of the start state, see Figure 8 in A.5.

In our experiments, we use the purely adversarial Free-For-All (FFA) environment. This environment is introduced in (MultiAgentLearning, 2018; Resnick et al., 2018) and we point the reader there for more details. The winner of the game is the last agent standing. It is played from the perspective of one agent whose starting position is uniformly picked among the four. The three opponents are

copies of the winner of the June 3rd 2018 FFA competition, a stochastic agent using a Finite State Machine Tree-Search approach (FSMTS, Zhou et al. (2018)). We also make use of the FSMTS agent as the 'expert' in the Backplay demonstrations.

The observation state is represented by 19 11x11 maps, and we feed the concatenated last two states to our agent as input. A detailed description of this mapping is given in A.5. The game ends either when the learning agent wins or dies, or when 800 steps have passed. Upon game end, the agent receives $+1$ for winning and $-1$ otherwise (**Sparse**). We also run experiments where the agent additionally receives $+0.1$ whenever it collects an item (**Dense**).

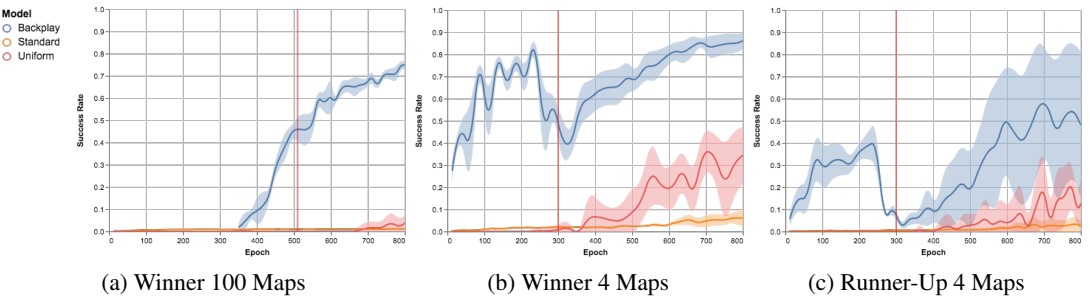

| (a) Winner 100 Maps | (b) Winner 4 Maps | (c) Runner-Up 4 Maps |

Figure 3. Pommerman results (5 seeds) when training with sparse rewards. Plots **a** and **b** are trained from the perspective of the winning agent, while **c** is trained from that of the runner up. The red bar indicates when the Backplay models begin training only on the initial state. Plot **a** is our starkest result and displays results on games starting from the initial state only regardless of when training occurs. you can see here that Backplay attains strong results where Uniform and Standard fail to learn anything of note.

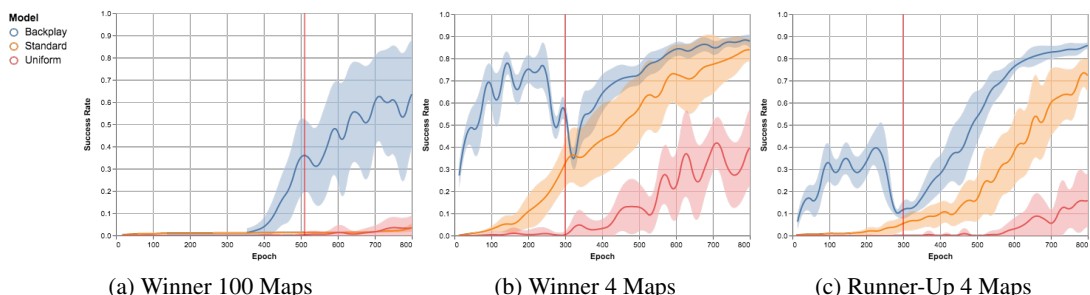

| (a) Winner 100 Maps | (b) Winner 4 Maps | (c) Runner-Up 4 Maps |

Figure 4. Pommerman results (5 seeds) when training with dense rewards. Again, **a** and **b** are trained from the perspective of the winning agent, while **c** is trained from that of the runner up. The cause of the higher variance in **a** was one of the seeds was worse than the others. Nonetheless, they all still did much better than either Standard or Uniform.

Our first three scenarios follow the Sparse setup. We independently consider three Backplay trajectories, one following the winner over 100 maps, one following the winner over four maps, and one following the runner up over four maps, with the latter two set of maps being the same. Figure 3 shows that **Backplay can soundly defeat the FSMTS agents in sparse settings when following the winner, even when there are 100 maps to consider, while other methods struggle in this setup**. Modulo higher variance in Backplay's result, we see a similar comparison in the runner-up case. Visit this link for an example gif of our trained Backplay agent (top left - red). Note that our agent learned to 'throw' bombs, a unique playing style that no prior Pommerman competitor had exhibited, including the FSMTS demonstrator.

Moreover, by training on 100 maps, Backplay generalizes (to some extent) on unseen boards. **Backplay wins 416 / 1000 games on a held out set of ten maps**, with the following success rates on each: 85.3%, 84.1%, 81.6%, 79.5%, 52.4%, 47.4%, 38.1%, 22.6%, 20%, and 18.3%. This was in contrast to our Maze experiments where no approach generalized. Given this discrepancy, we believe that the lack of generalization was a consequence of not including enough mazes during training.

## 5 CONCLUSION

We have introduced and analyzed Backplay, a technique which improves the sample efficiency of model-free RL by constructing a curriculum around a demonstration. We showed that Backplay agents can learn in complex environments where standard model-free RL fails, that they can outperform the 'expert' whose trajectories they use while training, and that they compare very favorably to related methods such as reversible curriculum generation. We also presented a theoretical analysis of its sample complexity gains in a simplified setting.

An important future direction is combining Backplay with more complex and complementary methods such as Monte Carlo Tree Search (MCTS, (Browne et al., 2012; Vodopivec et al., 2017)). There are many potential ways to do so, for example by using Backplay to warm-start MCTS.

Another direction is to use Backplay to accelerate self-play learning in zero-sum games. However, special care needs to be taken to avoid policy correlation during training (Lanctot et al., 2017) and thus to make sure that learned strategies are safe and not exploitable (Brown & Sandholm, 2017).

A third direction is towards non-zero sum games. It is well known that standard independent multi-agent learning does not produce agents that are able to cooperate in social dilemmas (Leibo et al., 2017; Lerer & Peysakhovich, 2017; Peysakhovich & Lerer, 2017; Foerster et al., 2017) or risky coordination games (Yoshida et al., 2008; Peysakhovich & Lerer, 2018). In contrast, humans are much better at finding these coordinating and cooperating equilibria (Bó, 2005; Kleiman-Weiner et al., 2016). Thus, we conjecture that human demonstrations can be combined with Backplay to construct agents that perform well in such situations.

Other future priorities are to gain further understanding into when Backplay works well, when it fails, and how we can make the procedure more efficient. Could we speed up Backplay by ascertaining confidence estimates of state values? Do the gains in sample complexity come from value estimation like our analysis suggests, from policy iteration, or from both? Is there an ideal rate for advancing the curriculum window and is there a better approach than a hand-tuned schedule?

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

# A  APPENDIX

## A.1  BACKPLAY HYPERPARAMETERS

As mentioned in Section 3, the Backplay hyperparameters are the window bounds and the frequency with which they are shifted. When we get to a training epoch represented in the sequence of epochs, we advance to the corresponding value in the sequence of windows. For example, consider training an agent with Backplay in the Maze environment (Table 3) and assume we are at epoch 1000. We will select a maze at random, an $N \in [16, 32)$, and start the agent in that game $N$ steps from the end. Whenever a pair is chosen such that the game's length is smaller than $N$, we use the initial state.

There isn't any downside to having the model continue training in a window for too long, albeit the ideal is that this method increases the speed of training. There is however a downside to advancing the window too quickly. A scenario common to effective training is improving success curves punctured by step drops whenever the window advances.

| Starting at training epoch | Uniform window |
|---|---|
| 0 | [0, 4) |
| 350 | [4, 8) |
| 700 | [8, 16) |
| 1050 | [16, 32) |
| 1400 | [32, 64) |
| 1750 | [64, 64) |

Table 3: Backplay hyperparameters for Maze.

| Starting at training epoch | Uniform window |
|---|---|
| 0 | [0, 32) |
| 50 | [24, 64) |
| 100 | [56, 128) |
| 150 | [120, 256) |
| 200 | [248, 512) |
| 250 | [504, 800) |
| 300 | [800, 800] |

Table 4: Backplay hyperparameters for Pommerman 4 maps.

| Starting at training epoch | Uniform window |
|---|---|
| 0 | [0, 32) |
| 85 | [24, 64) |
| 170 | [56, 128) |
| 255 | [120, 256) |
| 340 | [248, 512) |
| 425 | [504, 800) |
| 510 | [800, 800] |

Table 5: Backplay hyperparameters for Pommerman 100 maps.

## A.2  MAZE: DEMONSTRATION DETAILS

For N-Optimal demonstrations, we used a noisy A* where at each step, we follow A* with probability $p$ or choose a random action otherwise. We considered $N \in \{5, 10\}$. In all scenarios, we only selected maps in which there exists at least a path from the the initial state to the goal state, we filtered any path that was less than 35 in length and stopped when we found a hundred valid training games. Note that we held the demonstration length invariant rather than the optimal length (i.e. all N-optimal paths have the same length regardless of N, which means that the length of the optimal path of a N-optimal demonstration decreases with N). This could explain why the results in Table 2 (column 1) show that Backplay's performance increases with N (since the larger the N, the smaller the true optimal path, so the easier it is to learn an optimal policy for that maze configuration).

### A.3 MAZE: NETWORK ARCHITECTURE AND TRAINING PARAMETERS

We use a standard deep RL setup for our agents. The agent's policy and value functions are parameterized by a convolutional neural network with 2 layers each of 32 output channels, followed by two linear layers with 128 dimensions. Each of the layers are followed by ReLU activations. This body then feeds two heads, a scalar value function and a softmax policy function over the five actions. All of the CNN kernels are 3x3 with stride and padding of one.

We train our agent using Proximal Policy Optimization (PPO, Schulman et al. (2017)) with $\gamma = 0.99$, learning rate $1 \times 10^{-3}$, batch size 102400, 60 parallel workers, clipping parameter 0.2, generalized advantage estimation with $\tau = 0.95$, entropy coefficient 0.01, value loss coefficient 0.5, mini-batch size 5120, horizon 1707, and 4 PPO updates at each iteration. The number of interactions per epoch is equal to the batch size (102400).

The hyperparameters used for training the agent with Reverse Curriculum Generation (Florensa et al., 2017) are: $10^4$ rollout states for nearby sampling, 50 Brownian steps, 200 samples from new starts, 100 samples from old starts, interval for the expected return $[0.1, 0.9]$.

### A.4 MAZE: LEARNING CURVES

Below are our learning curves for the Maze challenge over five seeds. Note that we do not show any results for Standard as it failed to learn much of anything in the time allotted (3500 epochs). Also note that Backplay occasionally sees the actual grid starting position from epoch 1400, but it becomes the default starting state at epoch 1750. To align with this, our Florensa baseline switches to training from the actual start position at epoch 1750, and we show results for Uniform *only* over the initial starting state. Correspondingly, we show the graphs from epoch 1000 as all of the methods have commensurately poor results before that time.

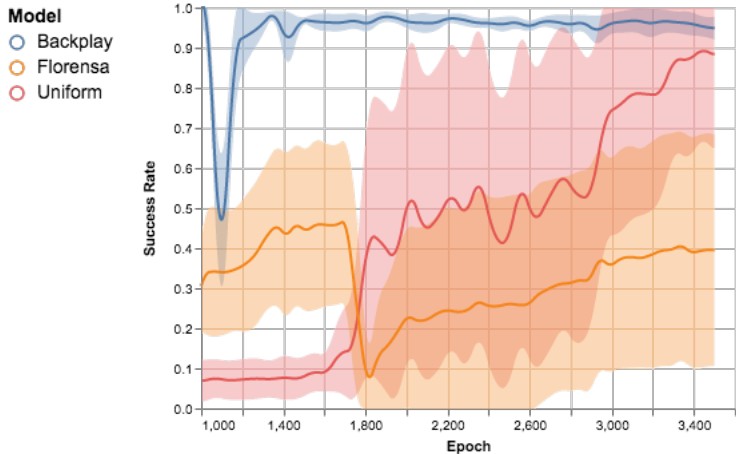

Figure 5. Maze results when training with demonstrations of length equal to the optimal path length. Note that Backplay has a very small variance and a very high success rate as early as epoch 1800. On the other hand, Florensa fails to break 70% and has a high variance, and Uniform doesn't achieve a strong success rate until epoch 3000.

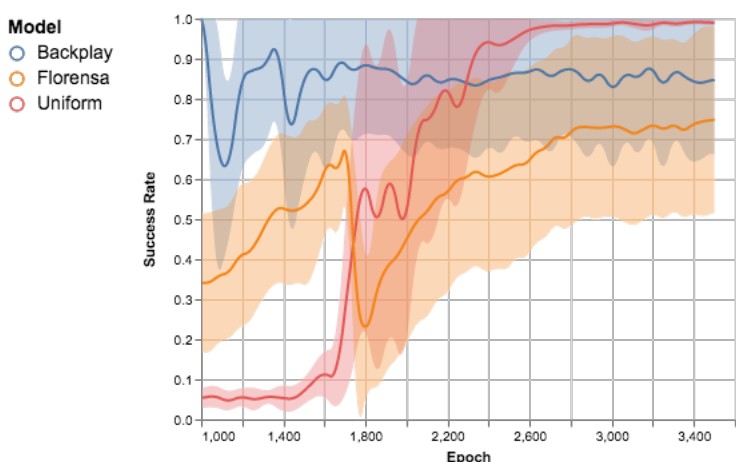

Figure 6. Maze results when training with demonstrations that are five steps longer than the optimal path length. Compared to the prior graph, Backplay doesn't do as well, albeit it still performs favorably compared to Florensa, with a consistently higher expected return. Its advantage over Uniform is a reduced amount of necessary samples.

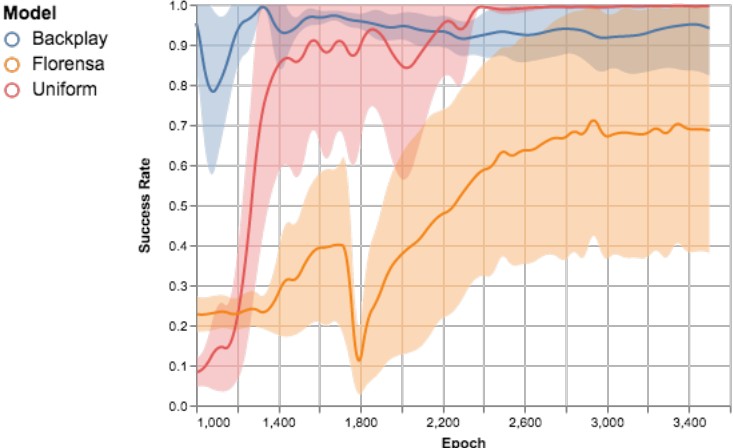

Figure 7. Maze results when training with demonstrations that are ten steps longer than the optimal path length. We again see that Backplay does very well compared to the Florensa baseline, with a much stronger expected return and lower variance. We also see, however, that Uniform is an able competitor to both of these as the expert becomes more suboptimal.

## A.5 Pommerman: Observation State

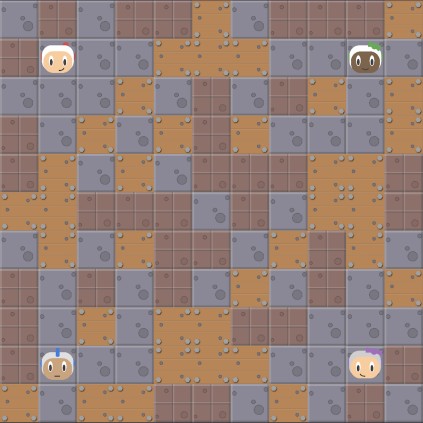

Figure 8. Pommerman start state. Each agent begins in one of four positions. Yellow squares are wood, brown are rigid, and gray are passages.

There are 19 feature maps that encompass each observation. They consist of the following: the agents' identities and locations, the locations of the walls, power-ups, and bombs, the bombs' blast strengths and remaining life counts, and the current time step.

The first map contains the integer values of each bomb's blast strength at the location of that bomb. The second map is similar but the integer value is the bomb's remaining life. At all other positions, the first two maps are zero. The next map is binary and contains a single one at the agent's location. If the agent is dead, this map is zero everywhere. The following two maps are similar. One is full with the agent's integer current bomb count, the other with its blast radius. We then have a full binary map that is one if the agent can kick and zero otherwise.

The next maps deal with the other agents. The first contains only ones if the agent has a teammate and zeros otherwise. This is useful for building agents that can play both team and solo matches. If the agent has a teammate, the next map is binary with a one at the teammate's location (and zero if she is not alive). Otherwise, the agent has three enemies, so the next map contains the position of the enemy that started in the diagonally opposed corner from the agent. The following two maps contain the positions of the other two enemies, which are present in both solo and team games.

We then include eight feature maps representing the respective locations of passages, rigid walls, wooden walls, flames, extra-bomb power-ups, increase-blast-strength power-ups, and kicking-ability power-ups. All are binary with ones at the corresponding locations.

Finally, we include a full map with the float ratio of the current step to the total number of steps. This information is useful for distinguishing among observation states that are seemingly very similar, but in reality are very different because the game has a fixed ending step where the agent receives negative reward for not winning.

## A.6 Pommerman: Network Architecture and Training Parameters

We use a similar setup to that used in the Maze game. The architecture differences are that we have an additional two convolutional layers at the beginning, use 256 output channels, and have output dimensions of 1024 and 512, respectively, for the linear layers. This architecture was not tuned at all during the course of our experiments. Further hyperparameter differences are that we used a learning rate of $3 \times 10^{-4}$ and a gamma of 1.0. These models trained for 72 hours, which is $\sim$50M frames. [2]

---

[2]In our early experiments, we also used a batch-size of 5120, which meant that the sampled transitions were much more correlated. While Backplay would train without any issues in this setup, Standard would only marginally learn. In an effort to improve our baselines, we did not explore this further.

## A.7 POMMERMAN: ACTION DISTRIBUTION ANALYSIS

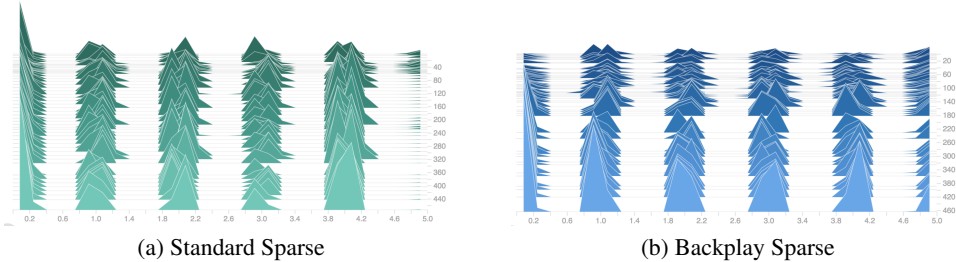

(a) Standard Sparse                    (b) Backplay Sparse

Figure 9. Typical histograms for how the Pommerman action selections change over time. From left to right are the concatenated counts of the actions (Pass, Up, Down, Left, Right, Bomb), delineated on the y-axis by the epoch. Note how the Standard agent learns to not use bombs.

Pommerman can be difficult for reinforcement learning agents. The agent must learn to effectively wield the bomb action in order to win against competent opponents. However, bombs destroy agents indiscriminately, so placing one without knowing how to retreat often results in negative reward for that agent. Since agents begin in an isolated area, they are prone to converging to policies which do not use the bomb action (as seen in the histograms in Figure 9), which leads them to sub-optimal policies in the long-term.

## A.8 POMMERMAN: WIN RATES

Here we show the per-map win rates obtained by the agent trained with Backlpay on the 100 Pommerman maps.

| Win | Maps |
|-----|------|
| 90% | 24 |
| 80% | 26 |
| 70% | 6 |
| 60% | 5 |
| 50% | 5 |

Table 6: Aggregate per-map win rates of the model trained with Backplay on 100 Pommerman maps. The model was run over 5000 times in total, with at least 32 times on each of the 100 maps. The *Maps* column shows the number of maps on which the Backplay agent had a success rate of at least the percent in the *Win* column. Note that this model has a win rate of $> 80\%$ on more than half of the maps and a win rate of $> 50\%$ on all maps.

## A.9 PRACTICAL FINDINGS

We trained a large number of models through our research into Backplay. Though these findings are tangential to our main points (and are mainly qualitative), we list some observations here that may be helpful for other researchers working with Backplay.

First, we found that Backplay does not perform well when the curriculum is advanced too quickly, however it does not fail when the curriculum is advanced 'too slowly.' Thus, researchers interested in using Backplay should err on the side of advancing the window too slowly rather than too quickly.

Second, we found that Backplay does not need to hit a high success rate before advancing the starting state window. Initially, we tried using adaptive approaches that advanced the window when the agent reached a certain success threshold. This worked but was too slow. Our hypothesis is that what is more important is that the agent gets sufficiently exposed to enough states to attain a reasonable barometer of their value rather than that the agent learns a perfectly optimal policy for a particular set of starting states.

Third, Backplay can still recover if success goes to zero. This surprising and infrequent result occurred at the juncture where the window moved back to the initial state and even if the policy's

entropy over actions became maximal. We are unsure what differentiates these models from the ones that did not recover.

We also explored using DAgger (Ross et al. (2011)) for training our agent, but found that it achieved approximately the same win rate ($\sim 20\%$) as what we would expect when four FSMTS agents played each other (given that there are also ties).

