# OpenReview forum: "Backplay: 'Man muss immer umkehren'"
_ICLR.cc/2019/Conference_

### Official Review · AnonReviewer1 · 2018-11-02
**Minor contributions; not a novel idea with limited evaluation**

**Rating:** 5
**Confidence:** 4

**Review:**

This paper presents a method for increasing the efficiency of sparse reward RL methods through a backward curriculum on expert demonstrations. The method in the paper is as follows: assuming access to expert demonstration and a resettable simulator, the start state of the agent in the beginning of training is sampled from end of demonstration (close to the rewarding state) where the task of achieving the goal is easy. Then gradually through a curriculum this is shifted backwards in the demonstration, making the task gradually harder.

The proposed method is closely related to 1) “Learning Montezuma’s Revenge from a Single Demonstration” a blog post and open-source code release by Salimans and Chen (OpenAI Blog, 2018) where they show that constructing a curriculum that gradually moves the starting state back from the end of a single demonstration to the beginning helps solve Montezuma’s revenge game 2) “Reverse Curriculum Generation for Reinforcement Learning” by Florensa et al. (CoRL 2017) , where they start the training to reach a goal from start states nearby a given goal state and gradually the agent is trained to solve the task from increasingly distant start states.

The approach is evaluated on a pair of tasks, a maze environment and a stochastic four-player game, Pommerman. In the maze environment, they compare to vanilla PPO and Uniformly sampled starting points across the expert trajectory. The Backplay method outperforms the vanilla baseline, however, from the training curves (~3500 epochs) in the appendix A4, it looks like the Uniform sampling baseline is doing as well or better than the proposed method. As pointed out by the authors themselves the reverse curriculum does not seem necessary in this environment. Also, it is unclear to me whether the curves shown is comparable as the starting point of the agent, at least in the beginning of training, is close to the goal with higher success rate for the Backplay method compared to baselines. A good convincing assessment would be to report success rate against the same starting point for all methods preferably not from the starting point of the demonstrations to assess generalisation of these methods for which authors briefly report unsuccessful results.

The Pommerman environment is more complex and the results reported are more interesting. Figure 3 shows the results on four different maps for which expert demonstrations are generated from a Finite-Machine Tree-search method (a competitive method in this environment). I’m slightly confused by the plots and the significant drops in performance once the curriculum is finished and agent encounters the start position of the demonstration trajectory (epoch 250). Is this affected by the schedule of the curriculum? Also, the choice of terminating training at epoch 550 is not clear as the method does not seem to have converged yet (the variance is quite high) and would be interesting to observe the dynamics of learning as the training proceeds and whether it converges to a stable policy at all. I am also slightly unclear regarding the performance difference between Standard method in Figure 3(c) and 3(d). If the Standard method is still the same baseline, vanilla PPO, why such huge performance difference? In my understanding, only the Uniform and Backplay methods should be affected by the quality of demonstrations? I believe this figure needs more explanation and clarity. I am also not clear on why Standard method is terminated at epoch 450 while other methods are trained until epoch 550. Figure 4 reports results of generalisation to 10 unseen maps but again the choice of terminating training after 550 epochs is not clear to me as the method again does not seem to have converged.

Overall, the choice of parameters is not well motivated, these include the window size for sampling the start point, the schedule for shifting the start point, batch size (102400 seems large to me and this choice is never explained), horizon (in appendix A3 reported to be 1707 for Maze while in the main text it is reported as 200 steps), termination of training (3500 for Maze, Figure 7, and 550 in Pommerman, Figure 3).

I commend the authors for honestly reporting their method’s shortcomings such as failure in generalisation, however, I find that the work lacks significance and quality. There is not much novelty in the proposed method and there is a clear lack of comparisons to existing sample efficient LfD techniques such as Generative Adversarial Imitation Learning (GAIL). I believe this paper requires substantial improvements for publication and is not up to the ICLR standards in its current form.

---

> ### Author Response · Authors · 2018-11-09
> **Response to reviewer AnonReviewer1: High Level**
>
> Thank you so much for your review. We split this response into two parts so that we could address the high level review and then the individual parts:
>
> "I find that the work lacks significance and quality. There is not much novelty in the proposed method and there is a clear lack of comparisons to existing sample efficient LfD techniques such as GAIL. I believe this paper requires substantial improvements for publication and is not up to the ICLR standards in its current form."
> - We would like for the reviewer to revisit this comment and their feeling that our work lacks significance and novelty:
>
> a. Novelty: Our approach is genuinely new. The closest comparison (OpenAI) debuted two weeks prior to ours and in the form of a blog post rather than a paper. The second closest technique (Florensa) required that the environment was both resettable and reversible (rather than just resettable) and they used random backward walks to generate the curriculum. These are not small differences. The Pommerman environment is not reversible - we cannot reverse the state past the point where an agent dies. Neither are other research environments like Starcraft or a robot manipulating breakable objects. Random backward walks are inefficient and use a lot of time exploring areas not useful for learning an optimal policy. However, in an effort to instill how impactful these differences are, we will run Florensa's technique on the Grid environment and report comparisons.
>
> b. Significance: We debuted a technique that (1) achieves a higher success rate over a vastly smaller complexity than standard RL (Figure 4), while generalizing to some extent to unseen maps (page 8). It does this (2) given even a suboptimal expert's state trajectory and (3) in sparse reward environments. Further, we (4) included unique theoretical analysis detailing why this works. We think that these contributions are unmatched and humbly request that you reevaluate our score in light of them.
>
> c. Comparisons: GAIL and most other LFD techniques are limited to being approximately as good as the expert demonstration. By relaxing the state assumption, our technique surpasses the expert’s performance as an upper bound. We did run experiments with DAgger and found that results were poor, even after further training with RL. This is briefly mentioned at the very end of the appendix. Would it be satisfactory to you if we run further experiments with DAgger and report those in the next iteration?
>
> Given your review, we believe that we didn't convey this story well enough. However, we also believe that our paper makes two important contributions that will be valuable to the community: (i) We demonstrate that Backplay is a robust, lightweight, and simple strategy to overcome sparse rewards in resettable environments; (ii) We provide the first mathematical framework for this line of research s.t. that we can analyze imitation learning under simple (albeit strong) assumptions and consequently laying out interesting future research directions.
>
> We hope that after we fix our presentation and address your concerns, you will be willing to reevaluate your score. Thanks again!

---

> > ### Comment · AnonReviewer1 · 2018-12-02
> > **Thanks for your reply**
> >
> > Thank you for your reply to my comments and modifying the paper to increase its clarity.
> > However, I still have concerns regarding the novelty and strength of experimental results. I have increased my score to 5 to reflect the improvements made to the current version.
> >
> > Regarding novelty, I want to clarify that although OpenAI's work is in the form of a blog post, it is a very detailed one with open-source code and the date of its publication precedes the ICLR submission date by 85 days (July 4th) and has already been cited by the authors in the first version of the paper.
> > Independent of that, as brought up by another reviewer, designing such manual curriculum is not particularly novel, and the experimental evidence does not help alleviate this concern.
> >
> > Hence, I would say that in this current form the work is not ready to be published in ICLR.

---

> ### Author Response · Authors · 2018-11-09
> **Response to reviewer AnonReviewer1: Specific points**
>
> “Also, it is unclear to me whether the curves shown is comparable as the starting point of the agent, at least in the beginning of training, is close to the goal with higher success rate for the Backplay method compared to baselines. A good convincing assessment would be to report success rate against the same starting point for all methods preferably not from the starting point of the demonstrations to assess generalisation of these methods for which authors briefly report unsuccessful results.”
> - Starting at epoch 1750, the success rate is *always* relative to the same starting point for all methods. Taking this into account, the much better sample complexity for Backplay is apparent. We will improve the way that we display these results. (Unfortunately, no method generalized in this scenario, and so comparing to other starting points is fruitless.)
>
> “I’m slightly confused by the plots and the significant drops in performance once the curriculum is finished and agent encounters the start position of the demonstration trajectory (epoch 250). Is this affected by the schedule of the curriculum?”
> - The significant drop in performance was due to the agent having to start from the initial state *all* of the time. You can see similar but smaller versions of this drop throughout the training procedure for Backplay. We will clarify this further.
>
> “Also, the choice of terminating training at epoch 550 is not clear as the method does not seem to have converged yet (the variance is quite high) and would be interesting to observe the dynamics of learning as the training proceeds and whether it converges to a stable policy at all.”
> - We gave all of the models the same *total training time* (three days) rather than keeping the number of epochs constant. Given that this was confusing, we will change this in the next version by continuing training to an invariant number of epochs.
>
> “I am also slightly unclear regarding the performance difference between Standard method in Figure 3(c) and 3(d). If the Standard method is still the same baseline, vanilla PPO, why such huge performance difference? In my understanding, only the Uniform and Backplay methods should be affected by the quality of demonstrations? I believe this figure needs more explanation and clarity.”
> - The difference between 3c and 3d relative to Standard is only in the starting position of the agent. The board positions are not symmetric, and thus some starting positions are more advantageous than others wrt passage layout and item positioning. It is likely the latter that causes these learning curve differences as the agent often learns utilize the kick bomb item and it could have trouble if there aren’t any nearby.
>
> “I am also not clear on why Standard method is terminated at epoch 450 while other methods are trained until epoch 550. Figure 4 reports results of generalisation to 10 unseen maps but again the choice of terminating training after 550 epochs is not clear to me as the method again does not seem to have converged.”
> - See above re keeping the total training time invariant.
>
> “Overall, the choice of parameters is not well motivated, these include the window size for sampling the start point, the schedule for shifting the start point, batch size (102400 seems large to me and this choice is never explained), horizon (in appendix A3 reported to be 1707 for Maze while in the main text it is reported as 200 steps), termination of training (3500 for Maze, Figure 7, and 550 in Pommerman, Figure 3).”
> - We will motivate the HPs more clearly in the next iteration. Briefly:
>   a. We discussed the window size / starting point in A.9, but feel that a thorough treatment of this is out of scope as our paper presents these results for the first time in the literature and the standalone field of hyperparameter search is vast.
>   b. We originally used a smaller batch size (see footnote on p16 in the appendix) and found that Standard suffered while Backplay worked no problem. In the interest of a more fair comparison, we increased the batch size until PPO was given sufficiently decorrelated samples to train well.
>   c. The horizon of 1707 is a hyperparameter in PPO. The number of allowed steps (200) before the episode terminates is a different concept.

---

### Official Review · AnonReviewer3 · 2018-11-03
**Request for some clarifications.**

**Rating:** 5
**Confidence:** 3

**Review:**

Thanks for your submission.

The  authors present a very elegant strategy of using Backplay, that learns a curriculum around a suboptimal demonstration. The authors show the technique reaches an upper bound on sample complexity especially in sparse reward environments. The strength of the paper is the ability to learn from even 10 sub-optimal demonstrator trajectory thereby achieving optimality in reaching the goal. The biggest limitation of the method as with other vanilla model free RL is the lack of generalization.

A bit more motivation on the simplified assumption that function approximation would have been better. Although, such a simplification seems to be a natural candidate to be upper bounded by the longest shortest path from v_0 to v_*; consideration of such simplicaton on the neighbourhood structure of the graph with respect to the maximum vertex degree seems to be missing or cliques. Although, the authors comment about the strong assumptions being made to aid the analysis.

The authors explain the analysis in a very precise and the analysis seems to work. Although, the part of the analysis where connections are drawn to the reciprocal spectral gap is not very clear.

The authors discuss the limitation of the analysis in the case of the binary tree, that follows from the arguments before.

It will be great to see a more systematic approach to deciding how fast/slow the window should be updated to unify some of the findings from the empirical experiments as that seems to affect the way the agent trains using Backplay.

---

> ### Author Response · Authors · 2018-11-08
> **Response to AnonReviewer3 "Request for some clarifications."**
>
> Thanks so much for your review.
>
> We appreciate your recognition that one strength of the Backplay paper is the ability to learn from the very sub-optimal expert in the Gridworld. This is true to an even larger extent in more complex environments, and we stress that the fact that Backplay lets the agent do much better than the expert is a huge strength compared to most current LFD approaches, including both DAgger and GAIL. Additionally, note that when we gave Backplay more trajectories (in the 100 map Pommerman case), it did generalize. This is specified on page 7, just before section 5.
>
> Can you please clarify your second paragraph? We are unsure what you mean.
>
> Thanks for the kind words about our mathematical analysis. We think that this is a keen and unique strength of this paper compared to anything similar in the literature. We will strive in the next version to clarify the connections to the spectral gap.
>
> With regards to discussing a more systematic approach to the window updates, while we do not expect the reviewers to read the Appendix in detail, we did include information about this in A.9. We will address your request in the next version by showing the results of further experiments and making this association more clear.

---

### Official Review · AnonReviewer4 · 2018-11-09
**sensible method, but limited novelty and evaluation is lacking**

**Rating:** 5
**Confidence:** 4

**Review:**

The paper presents a strategy for solving sparse reward tasks with RL by sampling initial states from demonstrations. The method, Backplay, starts by sampling states near the end of a demonstration trajectory, so that the agent will be initialized to states near the goal. As training progresses, the initial state distribution is incrementally shifted towards earlier steps in the demonstration, until the agent is trained starting from the original initial state. The authors further provide an analysis of the sample complexity of this method on a simple MDP. The method is demonstrated on a maze navigation task and a challenging game Pommerman.

The method is simple and sensible, but not particularly novel. As the authors pointed out, a very similar strategy for using demonstrations was previously presented in an OpenAI blogpost, Learning Montezuma’s Revenge from a Single Demonstration. However since that work was not published, it should not be held against this paper. That being said, sampling initial states from demonstrations is a tried-and-true strategy in RL, and the manually designed curriculum is also not particularly novel. Therefore the method is mainly a minor tweak to longstanding techniques. The paper has also acknowledges these previous works. As such, a more thorough evaluation with previous methods, such as those for automatic curriculum generation (e.g. Florensa et al. 2017 and Aytar et al. 2018) is vital, but is very much lacking in the current set of experiments.

This work can also benefit from a more diverse set of tasks to better evaluate the effectiveness of the method, and provide more insight on when such a strategy is beneficial. The experiments were conducted only on discrete grid world tasks, and additional experiments in continuous domains could be valuable. In the maze task, Backplay is not significantly better than uniform. Pommerman is a much more compelling task and shows more promising improvements from backflip. However, training seems to have been terminated fairly early, before the performance for most policies have converged. In particular, the standard dense policy in figure 3c seems to be doing pretty well, will it catch up to the backplay policy with more training? It is also pretty unexpected that the uniform policies are doing so poorly, worse than the standard policy for the Pommerman experiments. Do the authors have any intuition on why this might be the case?

In figure 3, what is the initial state distribution used to evaluate the various methods? Are all policies initialized to the original initial state of a task, or are initial states sampled from the demonstrations? Given the periodic drops in performance for the backplay policies, it appears that the initial states might be changing according the curriculum during evaluation. If that is the case, it might not be a fair comparison for the other policies, especially for the standard policies, which are trained under different initial states.

As detailed in the appendix, the sliding windows for the curriculum do not seem to have a lot of overlap. This might be a potentially problematic design decision, since the sudden change in the initial state distribution, may cause the policy to “forget” about strategies learned for previous initial states. Has the authors experimented with other more gradual transition strategies?

I think this paper in its current form does not yet meet the bar for ICLR. But this line of work could be a potentially promising direction. More thorough evaluation, better baselines, and more diverse tasks can help to strengthen this work. Further analysis on the effects of different initialization strategies could also make for a compelling contribution.

---

> ### Author Response · Authors · 2018-11-26
> **Response to "AnonReviewer4"**
>
> Thank you very much for your feedback. Apologies for the delayed response.
>
> “The method is simple and sensible, but not particularly novel.“
>
> We emphasize the novelty and significance of our approach compared to others in the literature in both the response to Reviewer 1 and in the revised manuscript (see the Intro and section 3.2). Please review those responses and consider reevaluating this view.
>
> “As such, a more thorough evaluation with previous methods, such as those for automatic curriculum generation (e.g. Florensa et al. 2017 and Aytar et al. 2018) is vital, but is very much lacking in the current set of experiments.”
>
> We have added comparisons with the Florensa et al. 2017 method on Maze and shown that Backplay achieves better sample complexity, overall performance, and less variance (see Table 2 and Figures 5, 6, 7). We cannot run these experiments on Pommerman because it is not reversible, a requirement for Reverse Curriculum Generation (but not for Backplay).
>
> “However, training seems to have been terminated fairly early, before the performance for most policies have converged”
>
>
> We ran all the models for longer (800 epochs) and show the results in the updated version (see Figures 3 and 4). Backplay maintains its advantage over the Uniform and Standard baselines, showing significant gains in the Sparse reward setting.
>
> “This work can also benefit from a more diverse set of tasks to better evaluate the effectiveness of the method, and provide more insight on when such a strategy is beneficial. The experiments were conducted only on discrete grid world tasks, and additional experiments in continuous domains could be valuable.”
>
> We would like to politely point out that even though our environments are discrete, they are still extremely challenging for state-of-the-art model-free reinforcement learning algorithms (PPO), which fail to learn any useful policies from sparse reward. We believe continuous v. discrete is not a differentiating factor in this case, given that Backplay with PPO can be easily applied to any continuous task in a similar fashion.
>
> We also provide analytical and qualitative insight on when this method can be expected to help (see section 3). While this analysis is for a discrete setup, similar intuitions apply for continuous spaces.
>
> Thus, formally analyzing and empirically evaluating Backplay in continuous spaces as different from discrete spaces is outside of the paper’s scope.
>
> “It is also pretty unexpected that the uniform policies are doing so poorly, worse than the standard policy for the Pommerman experiments. Do the authors have any intuition on this?”
>
> Thank you for bringing this up. We would like to politely point out that Uniform only performs worse than Standard in the Dense reward setting and this observation is not true for the Sparse reward and Maze scenarios. Thus, we believe this could be due to the reward shaping in combination with the distribution of starting states during training used by Uniform. Under Uniform, a winning agent might get less reward when starting later in the trajectory (since it has fewer opportunities to get the extra reward from collecting items). This can result in high variance gradients, which make it difficult to learn.
>
> “What is the initial state distribution used to evaluate the various methods? Are all policies initialized to the original initial state, or are initial states sampled from the demonstrations? <…> It appears that the initial states might be changing according the curriculum during evaluation. If that is the case, it might not be a fair comparison for the other policies, especially for the standard policies, which are trained under different initial states.”
>
> Figures 3 and 4 (in the updated paper) show the success rate for Backplay measured from the initial states used during training which are sampled from the demonstrations according to the curriculum. The success rate for Uniform and Standard is measured from the original initial state of the task. After the red line, Backplay is always initialized in the original initial state, so the curves to the right of the red line provide a fair comparison among the models (all initialized at the original initial state of the environment). We show the earlier part to demonstrate what Backplay training curves look like and to verify that the expected drops in training do in fact happen.
>
> “The sliding windows for the curriculum do not seem to have a lot of overlap. This might be a potentially problematic design decision, since the sudden change in the initial state distribution, may cause the policy to “forget” about strategies learned for previous initial states. Has the authors experimented with more gradual transition strategies?”
>
> We ran experiments with more gradual transitions but we did not find that to improve performance. We also did not encounter this “forgetting” issue in our current set-up.

---

> > ### Comment · AnonReviewer4 · 2018-12-08
> > **thank you for the response**
> >
> > Thank you for taking the time to include additional experiments and clarifications to the paper. The new pommerman experiments do provide a clearer comparison of the different methods. But I am still concerned about the novelty and strength of the technical contribution of this work. So my original score will remain for the time being.

---

> > > ### Author Response · Authors · 2018-12-08
> > > **On Novelty**
> > >
> > > Thanks for reviewing the new version. With regards to Novelty, which prior works are you thinking of when you consider our submission?

---

### Author Response · Authors · 2018-11-26
**New Paper Update**

We have uploaded a revised version of the paper taking into account your suggestions and would greatly appreciate you taking a renewed look at our contribution.

The major changes include:

- Clarified our contributions and the novelty of our work. See the Intro for details, but succinctly:
    1. We characterize analytically and qualitatively which environments Backplay will aid. We include a lot of detail in this section (with examples and guiding intuition) in order to provide context for readers to understand the distinguishing aspects among the different algorithms at play.
    2. We demonstrate Backplay's effectiveness on both a grid world task (to gain intuition) as well as the complex four player stochastic zero-sum game Pommerman.
    3. We empirically compare Backplay to other methods that improve sample complexity and show that our approach is (very) favorable.
- Clearly described the differences and advantages of Backplay compared to other methods (see Table 1).
- Restructured the theoretical analysis (see section 3.1) to be more approachable.
- Trained all the Pommerman models for longer and for a uniform amount (800 epochs). We think that this comparison is now much more fair and clearly shows the advantage in sample complexity when using Backplay (see Figures 3 and 4).
- Added comparisons on the Maze setup with Reverse Curriculum Generation approach as described in Florensa et al. 2017 (see Table 2, Figures 5, 6, 7). We cannot run these experiments on Pommerman given that it is not reversible, which is a requirement for Reverse Curriculum Generation (but not one for Backplay).
- Commented on the performance of behavioral cloning and other imitation learning algorithms on our tasks (see sections 4.1 and 4.2).

We also ran Backplay using a few different sets of curriculum hyperparameters (i.e. the window bounds and the frequency with which they are shifted) on the Maze setup. We did not find any significant difference in the results, so we decided to not include them in the paper for brevity. More details about the choice of hyperparameters and how they influence training can be found in the Appendix A.1.

We would like to thank, once more, to all reviewers for taking the time to provide valuable feedback that has certainly improved the quality of the manuscript.

---

### Meta-Review · Area_Chair1 · 2018-12-13
**Interesting paper but not quite there yet**

**Confidence:** 3
**Recommendation:** Reject

**Metareview:**


-pros:
- good, sensible idea
- good evaluations on the domains considered
- good analysis

-cons:
- novelty, broader evaluation

I think this is a good and interesting paper and I appreciate the authors' engagment with the reviewers.  I agree with the authors that it is not fair to compare their work to a blog post which hasn't been published and I have taken this into account.  However, there is still concern among the reviewers about the strength of the technical contribution and the decision was made not to accept for ICLR this year.